# Effective methods for the inactivation of *Francisella tularensis*

**Mika Azaki**[1,2], **Akihiko Uda**[1]*, **Deyu Tian**[3], **Katsuyoshi Nakazato**[2], **Akitoyo Hotta**[1], **Yasuhiro Kawai**[4], **Keita Ishijima**[1,5], **Yudai Kuroda**[1,5], **Ken Maeda**[1,5], **Shigeru Morikawa**[1]

**1** Department of Veterinary Science, National Institute of Infectious Diseases, Tokyo Japan, **2** Department of Correlative Study in Physics and Chemistry, Graduate School of Integrated Basic Sciences, Nihon University, Tokyo, Japan, **3** CAS Key Laboratory of Pathogenic Microbiology and Immunology, Institute of Microbiology, Chinese Academy of Sciences, Beijing, China, **4** Division of Biosafety Control and Research, National Institute of Infectious Diseases, Tokyo, Japan, **5** Laboratory of Veterinary Microbiology, Joint Faculty of Veterinary Medicine, Yamaguchi University, Yamaguchi, Japan

* auda@nih.go.jp

**Data Availability Statement:** All relevant data are within the paper and its Supporting Information file.

**Funding:** This study was partly supported by a grant-in-aid from the Ministry of Health, Labour and Welfare (H29-Shinko-Ippan-005) to SM, by a

## Abstract

*Francisella tularensis* (*F. tularensis*) is highly pathogenic to humans and must be handled under biosafety level 3 conditions. Samples used for the diagnosis and experimental analysis must be completely inactivated, although methods for the inactivation of *F. tularensis* are limited. In this study, effective methods for the inactivation of *F. tularensis* SCHU P9 and five other strains were determined by comparisons of colony-forming units between treated and control samples. The results showed that *F. tularensis* SCHU P9 was denatured by heat treatment (94°C for 3 min and 56°C for 30 min), filtration with a 0.22 μm filter, and the use of various solutions (i.e. >70% ethanol, methanol, acetone, and 4% paraformaldehyde). *F. tularensis* SCHU P9 remained viable after treatment with 50% ethanol for 1 min, filtration with a 0.45 μm filter, and treatments with detergents (i.e. 1% lithium dodecyl sulfate buffer, 1% Triton X-100 and 1% Nonidet P-40) at 4°C for 24 h. Additionally, *F. tularensis* SCHU P9 suspended in fetal bovine serum in plastic tubes was highly resistant to ultraviolet radiation compared to suspensions in water and chemically defined medium. The methods for inactivation of *F. tularensis* SCHU P9 was applicable to the other five strains of *F. tularensis*. The data presented in this study could be useful for the establishment of guidelines and standard operating procedures (SOP) to inactivate the contaminated samples in not only *F. tularensis* but also other bacteria.

## Introduction

Laboratory-acquired infections (LAIs) are caused by accidental exposure to infectious aerosols and contact with mucous membranes, even though LAIs have been decreased due to personal protective measures and biosafety training [1, 2]. Pike *et al.* reported that 4,079 LAIs resulting in 168 deaths occurred in the United States from 1930 to 1978 [3, 4]. Thereafter, Harding and Byers identified 1,267 LAIs resulting in 22 deaths [5]. According to Siengsanan-Lamont *et al.*, 27 LAIs occurred between 1982 and 2016 in the Asia-Pacific region [6]. In these instances, the

grant-aid from the Research Program on Emerging and Re-emerging Infectious Diseases from Japan Agency for Medical Research and development, AMED (grant no. 19fk0108097j0601) to AU, and by a grant-in-aid from the Japan Society for the Promotion of Science KAKENHI (16K09955) to AU.

**Competing interests:** The authors have declared that no competing interests exist.

LAIs were caused by *Brucella* spp., *Chlamydia psittaci*, *Coccidioides immitis*, *Coxiella burnetii*, *Francisella tularensis*, *Mycobacterium tuberculosis*, *Salmonella enterica*, *Blastomyces dermatitidis*, dengue virus, hepatitis B virus, and Venezuelan equine encephalitis virus. However, the causes in many cases of LAI have not been clearly identified in the past [7, 8].

Recently, the detailed causes of each LAI were investigated. In 2004, two laboratory workers at the National Institute of Virology Laboratory of the Chinese Center for Disease Control and Prevention (Beijing, China) were infected with severe acute respiratory syndrome coronavirus (SARS-CoV) due to the handling of the incompletely inactive virus [9]. It was thought that two individuals were the source of a subsequent small outbreak of SARS [9]. In 2014, the Centers for Disease Control and Prevention (Atlanta, GA, USA) announced that approximately 80 staff members had been exposed to *Bacillus anthracis* when samples were analyzed by matrix-assisted laser desorption ionization-time of flight mass spectrometry after transferring incompletely inactivated samples from a biosafety level (BSL)-3 facility to a lower BSL facility [10]. In 2004, three researchers at Boston University developed tularemia after accidental exposure to *F. tularensis* supposedly due to their failure to comply with safety protocols [11]. Considering these accidental infections with SARS-CoV, *B. anthracis* and *F. tularensis*, laboratory workers should pay careful attention when handling pathogens. Therefore, it seems necessary to educate staff members about methods for complete inactivation of pathogens.

Pathogenic microorganisms that pose a threat to public health are categorized into four risk groups according to the Laboratory Biosafety Manual published by the World Health Organization (Geneva, Switzerland) [12], the National Institutes of Health (NIH)–Office of Biotechnology Activities (Bethesda, MD, USA) publication titled 'The NIH Guidelines for Research Involving Recombinant or Synthetic Nucleic Acid Molecules' (NIH Guidelines) [13], 'Biosafety in Microbiological and Biomedical Laboratories Guide' (The U.S. Department of Health and Human Services, Centers for Disease Control and Prevention and NIH, USA) [14], and Directive 2000/54/EC of the European Parliament and of the Council of 18 September 2000 'on the protection of workers from risks related to exposure to biological agents at work' [15]. These guidelines obeyed at laboratories in the world are being revised based on historical incidents of accidental infection and the experience of the researchers and summarized by Kimman *et al.* [7]. In Japan, select agents and toxins are strictly classified according to the 'Act on the Prevention of Infectious Diseases and Medical Care for Patients with Infectious Diseases' adopted by the Ministry of Health, Labour and Welfare in 2007 [16]. All pathogens handled by the National Institute of Infectious Diseases (NIID) are classified into a risk group as determined by the Bio-Risk Committee and experiments conducted with risk group 3 pathogens must be performed in a BSL-3 facility in accordance with the regulations stipulated by the NIID, Japan. To prevent accidental infections, samples prepared at BSL-3 facilities are required to be completely inactivated if the samples are handled in an outside facility with the same or lower BSL.

*F. tularensis* is a Gram-negative facultative intracellular bacterium that is classified into four subspecies (subsp.): *tularensis*, *holarctica*, *mediasiatica*, and *novicida* [17]. Of these four subspecies, *F. tularensis* subsp. *tularensis*, which was first identified in North America, is the most pathogenic to both humans and animals, as the infectious dose is extremely low (<10 colony-forming units, CFU) [18, 19]. If untreated with antibiotics, the mortality rate is extremely high at around 30%–60% [18–20]. *F. tularensis* subsp. *holarctica* and *mediasiatica* have intermediate virulence and low mortality rates [21], whereas infection with subsp. *novicida* has only been detected in immunocompromised humans [21, 22]. In Japan, all *in vitro* and *in vivo* bacteriological procedures involving *F. tularensis*, with the exception of subsp. *holarctica* live vaccine strain (LVS), subsp. *tularensis* B38 strain and subsp. *novicida* must be conducted in a BSL-3 facility.

There are several established chemical and physiological techniques to inactive pathogens. Because *F. tularensis* does not form spores, inactivation is conducted with common methods, such as treatments with heat [23, 24], 4% paraformaldehyde (PFA) in phosphate-buffered saline (PBS) overnight [25], 4% PFA and 1% glutaraldehyde in 0.1M sodium cacodylate after formaldehyde [26], a combination of 10% sodium hypochlorite followed by 70% ethanol [27], and ultraviolet (UV) radiation [28]. *F. tularensis* subsp. *tularensis* SCHU S4 dried on acrylic, glass, polyamide, polyethylene, polypropylene, silicone rubber, and stainless steel was easily inactivated by exposure to vaporous hydrogen peroxide [29]. While, *F. tularensis* SCHU S4 dried on wood would be inactivated hardly by bleach, citric acid, 70% ethanol, quaternary Ammonia, and Pine-Sol [30]. Bone marrow-derived macrophages infected with *F. tularensis* SCHU S4 was completely fixed with 4% PFA for 5 min and 2% PFA for 15 min, whereas treatment with 1% PFA for 24 h failed to inactive infected cells [31]. However, the methods for the inactivation of *F. tularensis* differed among reports.

Therefore, the present study aimed to confirm the treatment conditions for the safe and complete inactivation of *F. tularensis* by comprehensive comparisons of the culturable bacteria between treated and control samples.

## Materials and methods

### Bacteria

*F. tularensis* subsp. *tularensis* SCHU P9 was established in a previous study [32] and cultured in chemically defined medium (CDM) at 37˚C until the late logarithmic phase. *F. tularensis* subsp. *tularensis* Nevada 14 and subsp. *holarctica* LVS, Kato, Yama, and Kf Water were kindly provided by Dr. H. Fujita (Ohara Research Laboratory, Ohara General Hospital, Fukushima, Japan) and listed in Table 1. They were cultured under the same conditions as *F. tularensis* SCHU P9. After centrifugation at $12,000 \times g$ for 2 min at 4˚C, bacterial pellets were resuspended in CDM containing 10% glycerol and stored at −80˚C until use. All procedures with regard to living bacterial cultures were performed in a BSL-3 facility in accordance with the regulations established by the NIID, Japan.

### Bacterial viability with short and long incubation periods

Five microliters of *F. tularensis* SCHU P9 (average, $1.0 \times 10^6$ CFU) was suspended in 100 μL of deionized water, CDM and undiluted fetal bovine serum (FBS) (Biosera, Nuaillé, France) and then incubated at 4˚C, 23˚C and 37˚C. After incubation for 0 min, 1 h, 1 day and 2, 4, 6, 8 and 10 weeks, serially diluted bacterial samples were cultured on Eugon chocolate agar plates at 37˚C for 4–7 days. The average CFU number in 100 μL was calculated from the average number of colonies of four replicated samples.

**Table 1. The list of *F. tularensis* strains used in this study.**

| Subspecies | Strain | Year of isolation | Location of isolation | Source | Biosafety level |
|---|---|---|---|---|---|
| subsp. *tularensis* | SCHU P9 | 2014 | Japan | subsp. *tularensis* SCHU | 3 |
| subsp. *tularensis* | Nevada 14 | 1953 | USA | hare | 3 |
| subsp. *holarctica* | LVS | 1961 | USA | Russian vaccine | 2 |
| subsp. *holarctica* | Kato | 1989 | Japan | human lymph node | 3 |
| subsp. *holarctica* | Yama | 1957 | Japan | Ixodes sp. | 3 |
| subsp. *holarctica* | Kf Water #23 | 1957 | USA | water | 3 |

## Inactivation by heat treatments

Five microliters of *F. tularensis* SCHU P9 (average, $5.2 \times 10^5$ CFU) suspended in 100 µL of deionized water, CDM, PBS (Sigma-Aldrich Corporation, St. Louis, MO, USA) or undiluted FBS was added to a 0.2 mL PCR tube (Bio-Bik Ina Optica, Nagano, Japan) and incubated at 94˚C and 56˚C using an Astec thermal cycler PC-806 (Astec Co., Ltd., Fukuoka, Japan). The control samples were incubated at 4˚C. After the indicated incubation time, the serially diluted bacterial samples were cultured on Eugon chocolate agar plates at 37˚C for 4–7 days. The average CFU number in 100 µL was calculated from the average number of colonies of four replicated samples.

## Bacterial counts before and after filtration

Fifty microliters of *F. tularensis* SCHU P9 (average, $2.9 \times 10^6$ CFU/ 100 µL) suspended in 1 mL of CDM was screened through Millipore PVDF Hydrophilic Millex-HV Sterile Syringe Filter Unit 0.45 Micron (SLHV033RS, EMD Millipore Corporation, Billerica, MA, USA) and 0.22 Micron (SLGV033RS, EMD Millipore Corporation). Before and after filtration, the samples were serially diluted and cultured on Eugon chocolate agar plates at 37˚C for 4–7 days. The average CFU number in 100 µL was calculated from the average number of colonies of four replicated samples.

## CFU in supernatants and pellets after centrifugation

Five microliters of *F. tularensis* SCHU P9 (average, $2.2 \times 10^6$ CFU) suspended in 100 µL of CDM was centrifuged at $12,000 \times g$ for 2 min at 4˚C, and the supernatant was transferred into a new tube. The remaining pellets were resuspended in 100 µL of CDM. The samples were serially diluted and cultured on Eugon chocolate agar plates at 37˚C for 4–7 days. The average CFU number in 100 µL was calculated from the average number of colonies of four replicated samples.

## Bacterial CFU after the inactivation using various solvents

Five microliters of *F. tularensis* SCHU P9 was mixed with 100 µL of deionized water, 10%–90% ethanol (Sigma-Aldrich), 100% methanol (Nacalai Tesque, Inc., Kyoto, Japan), 100% acetone (Sigma-Aldrich), a mixture of 50% methanol and 50% acetone, 10% formaldehyde neutral buffer solution (Wako Pure Chemical Industries, Ltd., Osaka, Japan), 4% PFA (Wako), 100% acetonitrile (Sigma-Aldrich) and final concentration of 0.001%–1% sodium hypochlorite (Purelux; Oyalox Co., Ltd., Tokyo, Japan), then incubated for 10 min at room temperature (23˚C).

Five microliters of *F. tularensis* SCHU P9 was mixed with 100 µL of 1% Triton X-100, 1% NP-40 and 1% LDS (Nacalai Tesque) buffer supplemented with $1 \times$ sodium dodecyl sulphate buffer, 10% glycerol (Wako) and 0.005% bromophenol blue (63 mM Tris-HCl, pH 6.8; Wako). These mixtures of detergents and bacteria were incubated at 4˚C for 10 min, 1 h, and 24 h. All samples were centrifuged at $12,000 \times g$ for 2 min at 4˚C. Afterward, the supernatant was discarded to remove the effects of solvents. After the bacterial pellets were resuspended in 100 µL of CDM, the viable bacteria were counted. These experiments were conducted using four replicates.

## Mechanical disruption using beads

Fifty microliters of *F. tularensis* SCHU P9 was added to 1 mL of CDM, undiluted FBS or Roswell Park Memorial Institute (RPMI) 1640 medium containing 10% FBS with and without

detergents. Bacterial suspensions were put into 2 mL tubes containing 6.35 mm ceramic spheres (MP Biomedicals, Illkirch-Graffenstaden, France) and Lysing Matrix A (garnet) (MP Biomedicals), shaken for 30 s periods at 4,200 rpm in a Mini Bead Beater (BioSpec Products, Inc., Bartlesville, OK, USA) and then immediately cooled on ice. Live bacteria in 100 μL aliquots were enumerated. These experiments were conducted using four replicates.

## The viability of *F. tularensis* SCHU P9 after treatments with commercial products

The inactive effect of Cell Lysis Buffer (10×) (Cell Signaling Technology, Danvers, MA, USA) was examined. Ninety microliters of *F. tularensis* SCHU P9 (average, $3.5 \times 10^6$ CFU) were mixed with ten microliters of Cell Lysis Buffer (10×) (Cell Signaling Technology), while CDM was added into bacteria as the control sample. Samples were incubated at 4˚C for 10 to 60 min.

Bacterial viability suspended in RLT buffer supplied in RNeasy Mini Kit was evaluated. Bacterial pellets (average, $1.1 \times 10^6$ CFU) after the centrifugation at $12,000 \times g$ for 2 min at 4˚C were suspended in 100 μL of RLT buffer alone, the mixture of an equal volume of RLT buffer and 70% ethanol, and CDM as the control sample. Samples were incubated at room temperature for 10 min.

In these experiments, samples after incubation were centrifuged at $12,000 \times g$ for 2 min at 4˚C to remove the commercial buffers. The pellets containing live bacteria were suspended in CDM and the CFU was counted. These experiments were conducted using four replicates.

## Bacterial viability after UV radiation

Five microliters of *F. tularensis* SCHU P9 spiked in 100 μL of deionized water, CDM or undiluted FBS were transferred into 1.5 mL tubes (Sarstedt K.K., Tokyo, Japan) and 0.2 mL PCR tubes (Bio-Bik Ina Optica). These samples were radiated at room temperature (23˚C) using a Funa-UV-Linker (FS-800; Funakoshi, Tokyo, Japan) equipped with a low-pressure lamp (254 nm). The average UV energy (3 mW/cm$^2$) was monitored with the sensor of this system during this experiment. After radiation, the CFU in 100 μL of these samples was measured. These experiments were conducted using four replicates.

## Bacterial CFU after the treatments in five strains of *F. tularensis*

Five *F. tularensis* strains of subsp. *tularensis* Nevada 14 and subsp. *holarctica* LVS, Kato, Yama, and Kf Water were prepared to validate the effective inactivation of *F. tularensis*. Bacteria were heated at 94˚C for 3 min and 56˚C for 30 min, filtered using Millipore PVDF Hydrophilic Millex-HV Sterile Syringe Filter Unit 0.45 Micron (SLHV033RS, EMD Millipore Corporation) and 0.22 Micron (SLGV033RS, EMD Millipore Corporation), incubated at 4˚C for 24 h with detergents (1% LDS buffer, 1% NP-40, and 1% TritonX-100), and radiated at room temperature (23˚C) with a low-pressure lamp (254 nm). These procedures were in accordance with the same conditions as those for the treatment of *F. tularensis* SCHU P9. The CFU of untreated and treated samples were compared. The experiments were conducted in four replicates.

## Statistical analysis

All statistical analyses were performed using GraphPad Prism v5 software (GraphPad Software, Inc., La Jolla, CA, USA). All experiments in this study were performed using four replicates. The results are expressed as the mean number of CFU ± SD. Significant differences in CFU between the heat-denatured samples and control samples were determined using the Student's *t*-test. Other comparisons of CFU between the treated and control samples were

performed using one-way or two-way analysis of variance (ANOVA). When significant differences were found, further comparisons were made using the Bonferroni *post hoc* test.

## Results

### Bacterial viability in deionized water, chemically defined medium (CDM), and undiluted fetal bovine serum (FBS) after different incubation periods

To evaluate the stability in *F. tularensis* SCHU P9, bacteria suspended in deionized water, undiluted FBS, and chemically defined medium (CDM) for *F. tularensis* were incubated at 4˚C, 23˚C and 37˚C, respectively. In deionized water, the number of live bacteria was approximately $1.2 \times 10^6$ CFU at 0 min, which was not significantly different from that at 1 h ($p > 0.05$). Thereafter, no bacteria incubated at 4˚C for 10 weeks (w), 23˚C for 2 weeks or 37˚C for 1 day (d) were detected (Fig 1A). CDM, which is often used for liquid culture of *F. tularensis*, promoted long-term bacterial viability at 4˚C to 37˚C (Fig 1B). In *F. tularensis* SCHU P9 suspended in undiluted FBS, bacterial viability at 4˚C, 23˚C and 37˚C was slightly improved than in deionized water (Fig 1C). Bacterial viability of this strain was not affected by short incubation of less than 1 h because no significant change in the number of CFU was detected between 0 min and 1 h ($p > 0.05$).

### Effective inactivation for *F. tularensis* SCHU P9

Heat treatment has been widely used for the complete inactivation of bacteria. Hence, the thermal resistance of *F. tularensis* SCHU P9 was examined in this study. Bacterial suspensions in deionized water, CDM and undiluted FBS containing approximately $5.2 \times 10^5$ CFU/100 μL of *F. tularensis* SHCU P9 were incubated at 94˚C for 3 min (Fig 2A) and 56˚C for 30 min (Fig 2B). In contrast, the control samples were incubated at 4˚C for the same time as for the heat treatment. The results revealed that heat treatment at 94˚C for 3 min and 56˚C for 30 min had completely inactivated *F. tularensis* SHCU P9, as no live bacteria were detected in any of the heat-treated samples. The minimal time required for heat inactivation of *F. tularensis* SHCU P9 was estimated. No viable bacteria were detected after incubation at 94˚C for 45 s (Fig 2C). As shown in Fig 2D, some live bacteria were detected in CDM (0.8 CFU) and undiluted FBS (0.5 CFU) following heat treatment at 56˚C for 5 min, but none that were suspended in deionized water and PBS. No bacteria were detected in heat-treated samples at 56˚C for 10 min (S1 Fig.).

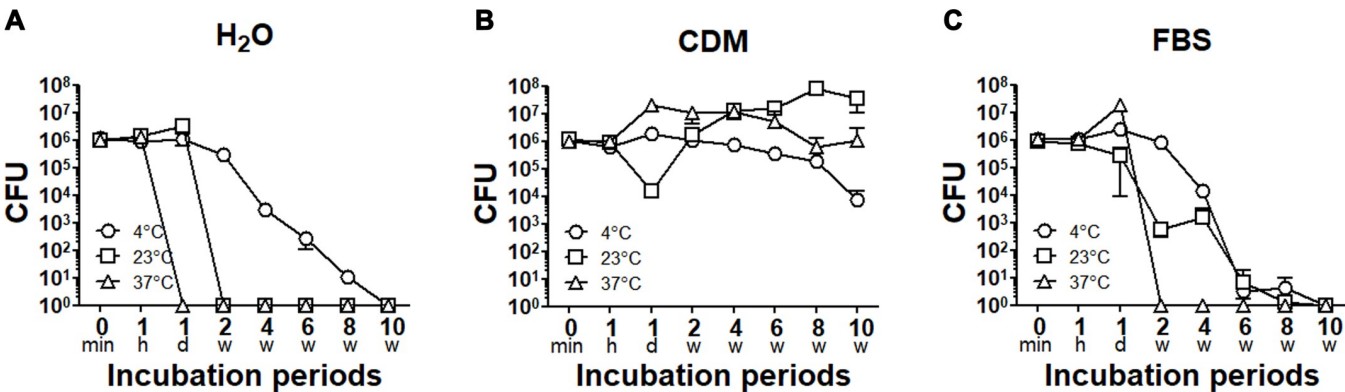

**Fig 1. Changes to the viability of *F. tularensis* SCHU P9 by long-term incubation.** Bacteria inoculated into deionized water (A), CDM (B) and undiluted FBS (C) were maintained at 4˚C (∘), 23˚C (□) and 37˚C (△). After incubation for 0 min, 1 h, 1 day (d) and 2, 4, 6, 8 and 10 weeks (w), the CFU numbers of four replicates of each bacterial sample were counted. The mean CFU ± standard deviations (SD) are shown.

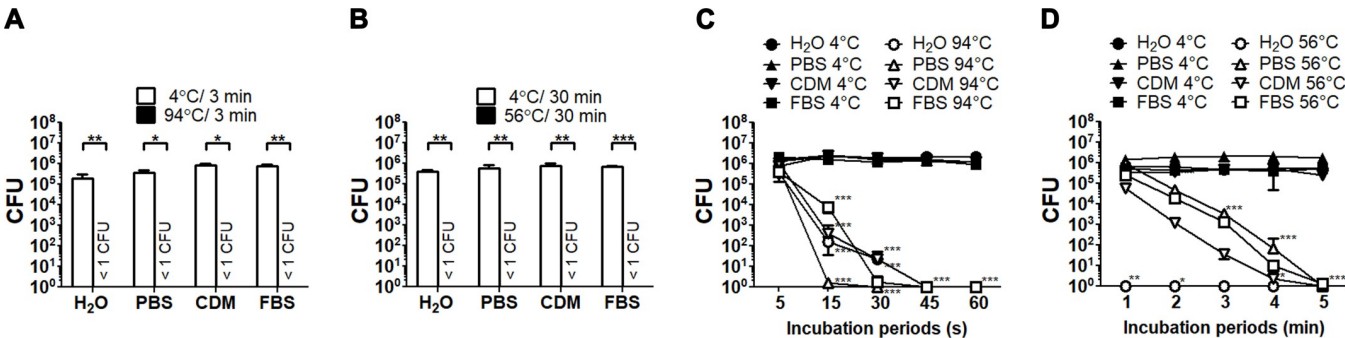

**Fig 2. Heat inactivation of *F. tularensis* SCHU P9.** Bacterial suspensions were prepared with deionized water, CDM, PBS, and undiluted FBS. (A and B) The samples were heated at 94°C for 3 min (A) and 56°C for 30 min (B) and then immediately cooled on ice. The control samples were cooled to 4°C for 3 min (A) and 30 min (B). The black and white bars indicate the CFU numbers of the treated and control samples, respectively. Statistical significance was determined by one-way ANOVA with a *post hoc* test (*$p < 0.05$, **$p < 0.01$ and ***$p < 0.001$). (C and D) Bacterial suspensions heated at 94°C (C) and 56°C (D) for the indicating times in the figure were immediately cooled on ice. In contrast, the control samples were cooled on ice for identical time periods. The white and black symbols indicate the CFU numbers of the treated and control samples, respectively. In all experiments, the CFU number of four replicates of each bacterial sample was counted. The mean CFU ±SD are shown. Statistical significance was determined by two-way ANOVA with a *post hoc* test (*$p < 0.05$, **$p < 0.01$ and ***$p < 0.001$).

A sterilizing filter is often used to remove contaminating bacteria from liquid samples. To evaluate the effectiveness of filter sterilization of *F. tularensis* SCHU P9, 1 mL of bacterial suspensions in CDM (average, $2.9 \times 10^6$ CFU/100 μL) were filtrated using a sterilizing filter with 0.22 (SLGV033RS, EMD Millipore Corporation) and 0.45 μm pores (SLHV033RS, EMD Millipore Corporation) (Fig 3A). The 0.22 μm filter removed all viable bacteria. On the other hand, a small number of bacteria (average, 5.5 CFU/100 μL) was detected in the filtrates through 0.45 μm filter, although the CFU number was significantly decreased ($p < 0.001$).

Using a solution of 70% ethanol is a simple aseptic technique to inactivate pathogens. Hence, the relationships between ethanol concentration and the viability of *F. tularensis* SCHU P9 were investigated. Bacteria suspended in 0%–90% ethanol were incubated for 10 min at room temperature (23°C) and then centrifuged to discard the ethanol solution. Afterward, the CFU number of pellets was calculated. The preliminary data showed no significant change in CFU number from before centrifugation (Fig 3B). After treatment with >50% ethanol for 10 min, no viable bacteria were detected (Fig 3C). On the other hand, the CFU numbers after treatment with 0%, 10% and 30% ethanol for 10 min were $3.1 \times 10^6$, $1.6 \times 10^6$ and $1.2 \times 10^4$, respectively (Fig 3C). Furthermore, the incubation time required for inactivation with the use of various concentrations of ethanol was determined. Bacteria suspended in 90% and 70% ethanol were rapidly inactivated within 15 s (Fig 3D), whereas those in 0%–50% ethanol had survived for incubation periods for 60 s, although the CFU number was significantly decreased by treatment with 50% ethanol ($p < 0.001$).

Organic solvents, such as formalin, methanol, and acetone are commonly used to fix infected tissue samples. The viable number of *F. tularensis* SCHU P9 following treatment with various fixation solutions was determined. Bacteria spiked into 10% formalin neutral buffer solution, 4% PFA, 100% methanol, 100% acetone, a mixture of 50% methanol/50% acetone and 100% acetonitrile were incubated for 10 min at room temperature. After the incubation periods, the samples were centrifuged to discard the solvents. The viable bacteria in pellets resuspended in CDM were counted. As shown in Fig 3E, there were no viable bacteria in any of the samples treated with organic solvents, whereas abundant bacteria (average, $1.2 \times 10^6$ CFU) were found in the control samples. The effective concentration of sodium hypochlorite to inactivate *F. tularensis* SCHU P9 was also evaluated. The results showed that the final concentration of 0.1% sodium hypochlorite had sufficiently inactivated the bacteria, and there

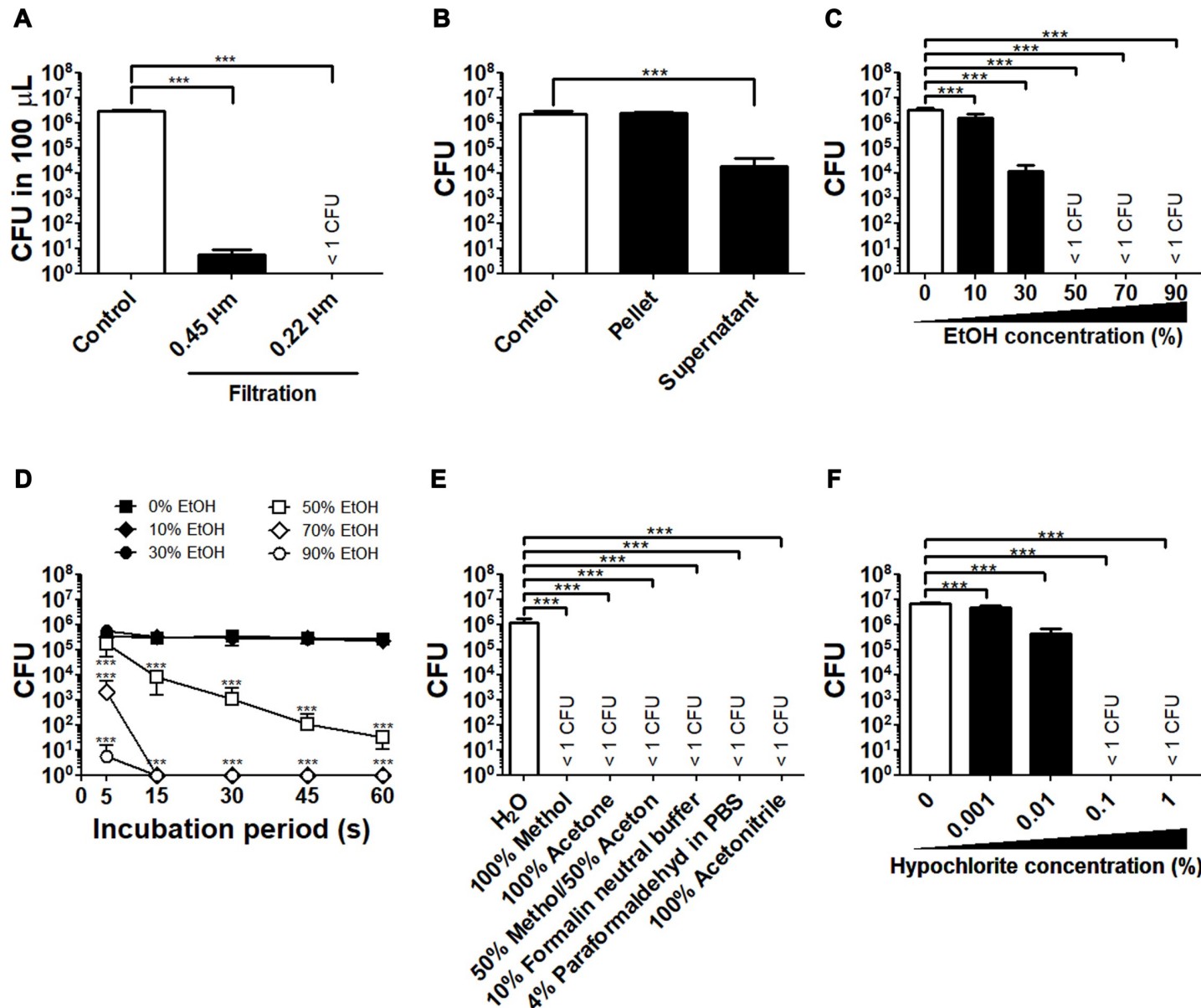

**Fig 3. The determinants of effective inactivation of *F. tularensis* SCHU P9.** (A) Bacterial suspensions in 1 mL of CDM were filtrated through 0.45 and 0.22 μm pore size membrane filters (Merck Millipore). The mean CFU ± SD in 100 μL before and after filtration are shown. (B) Bacterial suspensions in CDM were centrifuged at 12,000 × *g* for 2 min at 4˚C. The mean CFU ± SD before and after centrifugation is shown. (C to F) Bacteria suspended in 0%–90% ethanol (C and D), various solvents (E) and the final concentration of 0%–1% sodium hypochlorite (F) were incubated at room temperature for 10 min (C, E, and F) and 0–60 s (D). All experiments were performed using four replicates. The mean CFU ± SD are shown. Statistical significance was determined by one-way ANOVA (A, B, C, E and F) and two-way ANOVA (D)with the *post hoc* test (*$p < 0.05$, **$p < 0.01$ and ***$p < 0.001$).

were significant differences in bacterial viability between the treated and control samples ($p < 0.001$).

## Inactivation efficiency of detergents and UV radiation

Detergents are used widely for protein extraction from bacteria and infected cells. In this study, the numbers of live bacteria between detergent-treated and control samples were compared. Combined condition analysis was performed using three incubation periods (10 min, 1

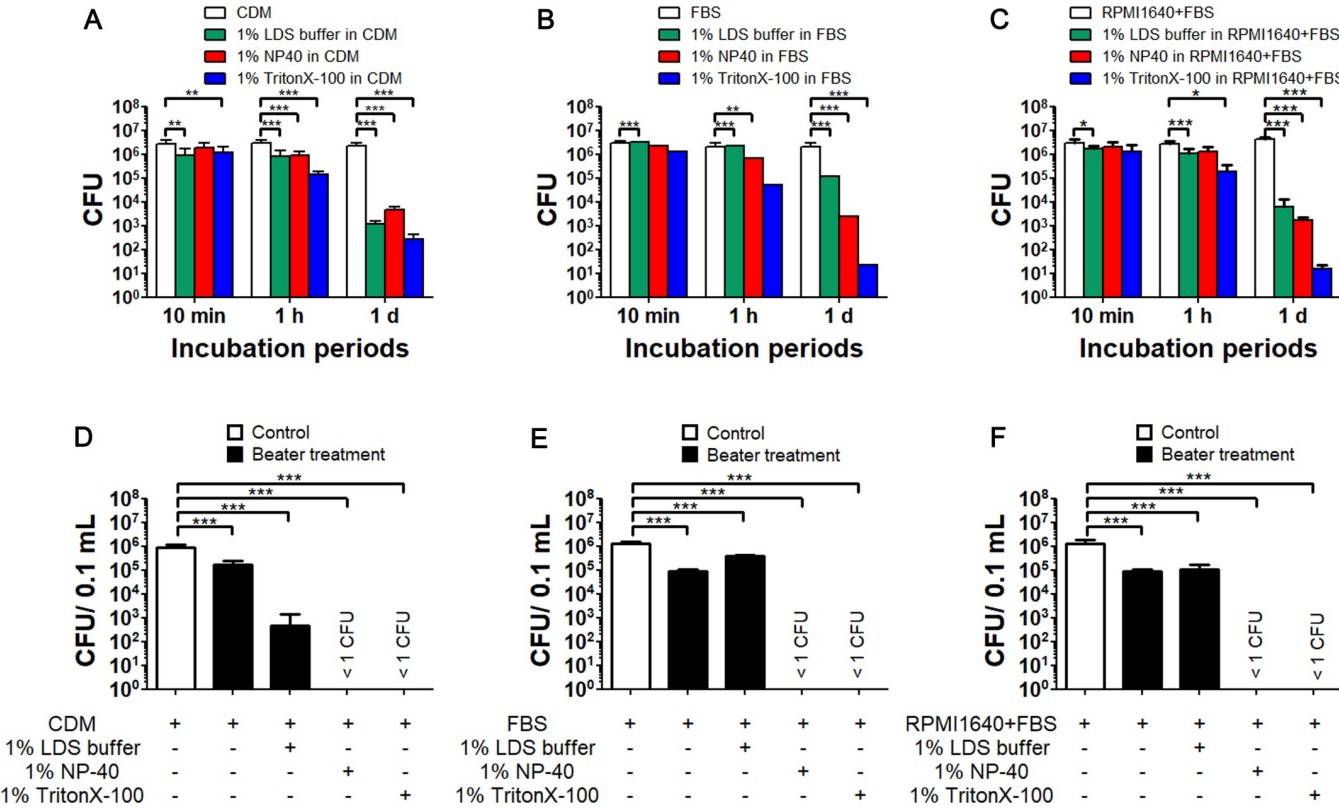

**Fig 4. Changes in the viability of *F. tularensis* SCHU P9 using detergents.** (A–C) Bacteria suspended in CDM (A), undiluted FBS (B) and RPMI 1640 containing 10% FBS (C) with and without detergents (1% LDS buffer, 1% NP-40 and 1% TritonX-100) were incubated at 4°C for 10 min, 1 h and 1 day. After incubation, the bacteria were immediately centrifuged, and the CFU number of the pellets was calculated. (D–F) Detergents (1% LDS buffer, 1% NP-40 and 1% TritonX-100) were added to bacterial suspensions in CDM (D), undiluted FBS (E) and RPMI 1640 containing 10% FBS (F). The samples were homogenized at 4,200 rpm for 30 s and then immediately cooled on ice. The mean CFU number ± SD of four replicates are shown. Statistical significance was determined by two-way ANOVA (A–C) and one-way ANOVA(D–F) with a *post hoc* test (***$p < 0.001$).

h, and 24 h), three different detergents (1% lithium dodecyl sulphate [LDS] buffer, 1% Nonidet P-40 [NP-40] and 1% Triton X-100) and three solvents (deionised water, CDM and RPMI 1640 containing 10% FBS) at 4°C. After incubation, all samples were centrifuged to remove the detergent solution. As shown by the mean CFU numbers ± standard deviation (SD) presented in Fig 4A–4C, no sample was completely inactivated, although CFU numbers in samples treated with detergents for 24 h were significantly lower than in the control samples ($p < 0.001$).

The bacterial samples containing the detergents were subjected to mechanical disruption using beads. Bacteria suspended in CDM, undiluted FBS and RPMI1640 containing 10% FBS were homogenated, and the mean CFU numbers ± SD were determined. As shown in Fig 4D–4F, the average CFU number before homogenisation in CDM, undiluted FBS and RPMI 1640 containing 10% FBS were $0.9 \times 10^6$ CFU/0.1 mL, $1.3 \times 10^6$ CFU/0.1 mL and $1.2 \times 10^6$ CFU/0.1 mL, respectively. Bacterial numbers in samples without detergent were reduced by approximately 1/10 by mechanical disruption. After mechanical disruption, live bacteria were detected in samples prepared with 1% LDS buffer but not those containing 1% NP-40 and 1% TritonX-100.

The viability of *F. tularensis* SCHU P9 after treatment with commercial products was evaluated. Bacterial pellets ($3.5 \times 10^6$ CFU) suspended in Cell Lysis Buffer (Cell Signaling Technology, Danvers, MA, USA) and the RLT buffer of the RNeasy mini kit (Qiagen Ltd., Valencia,

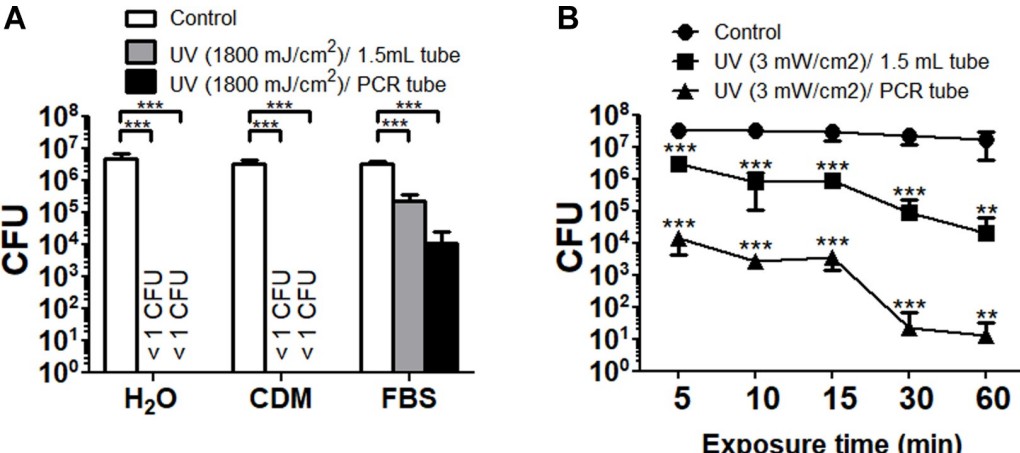

**Fig 5. The viability of *F. tularensis* SCHU P9 after treatments of commercial products.** Bacterial viability was evaluated after the treatment using Cell Lysis Buffer (Cell Signaling Technology) and the RLT buffer supplied by RNeasy mini kit (Qiagen Ltd.,). (A) Bacteria suspended in Cell Lysis Buffer and CDM (control) were incubated at 4°C for the indicated time. (B) Bacterial pellets after the centrifugation at 12,000 × g for 2 min at 4°C were suspended in RLT buffer alone, the mixture of RLT buffer and 70% ethanol, and CDM (control). The samples were incubated at room temperature for 10 min. All incubated samples were centrifuged at 12,000 × g for 2 min at 4°C and the pellets were suspended in CDM. the mean CFU ± SD of control and the treatment samples are shown. Statistical significance was determined by two-way ANOVA (A) and one-way ANOVA (B) with the *post hoc* test (***$p < 0.001$).

CA, USA) were incubated for 10 min at room temperature. After incubation, there was an abundance of bacteria in the samples treated with Cell Lysis Buffer but none in the samples treated with the RLT buffer (Fig 5).

The UV light is reported to inactivate bacterial cells via thymine dimer formation [33]. The reduction in viable *F. tularensis* SCHU P9 with UV radiation is shown in Fig 6A. In deionized water and CDM, no live bacteria were detected after UV radiation (1800 mJ/cm²) at room temperature. In undiluted FBS, the numbers of viable bacteria in both the 1.5 mL and 0.2 mL

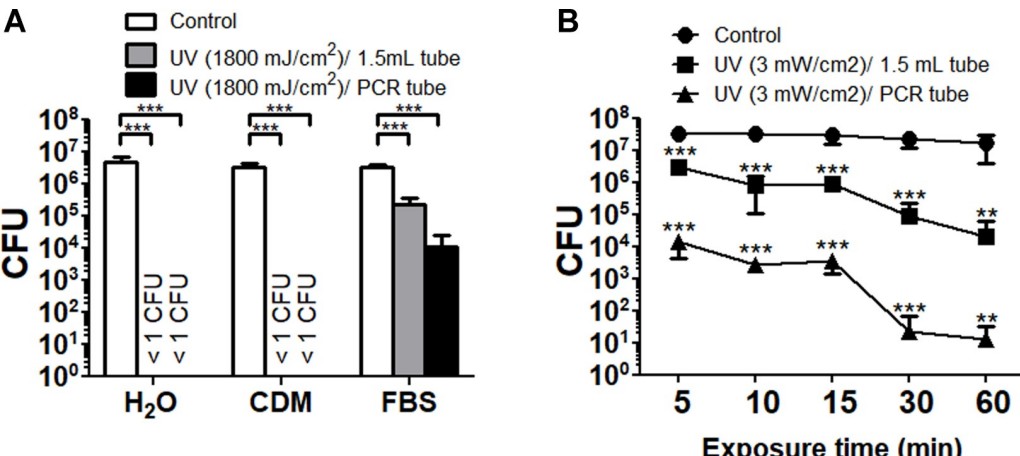

**Fig 6. Changes to the viability of *F. tularensis* SCHU P9 by UV radiation.** (A) Bacteria suspended in deionized water, CDM and undiluted FBS were aliquoted into four 1.5 mL tubes and 0.2 mL PCR tubes. These samples were simultaneously radiated with UV light (3 mW/cm² × 600 s = 1800 mJ/cm²) at 254 nm using FUNA-UV-LINKER FS-800. (B) Bacteria suspended in undiluted FBS were prepared in 1.5 mL tubes and 0.2 mL PCR tubes. After treatment for 5–60 min at 3 mW/cm², the CFU numbers of these samples were calculated. The mean CFU ± SD of four replicates are shown. Statistical significance was determined by two-way ANOVA with a *post hoc* test (**$p < 0.01$ and ***$p < 0.001$).

tubes were significantly decreased after UV radiation, but complete inactivation in *F. tularensis* SCHU P9 was not achieved (Fig 6A). Similarly, bacteria in undiluted FBS had survived after UV radiation at room temperature for 60 min (Fig 6B). On the other hand, *F. tularensis* SCHU P9 cells spread onto Eugon chocolate agar were completely inactivated by UV radiation at 6.3 mJ/cm$^2$.

## Viabilities after the various treatments in five strains of *F. tularensis*

It was evaluated whether the methods for inactivation of *F. tularensis* SCHU P9 described in this study would be applicable to other strains of *Francisella*. The CFU between treated and control samples were compared using five strains (subsp. *tularensis* Nevada 14 and subsp. *holarctica* LVS, Kato, Yama, and Kf Water). The results are shown in Fig 7. No live bacteria were detected in samples after the treatments with 94˚C for 3 min (Fig 7A black bars), 56˚C for 30 min (Fig 7A gray bars), 0.22 μm filtration (Fig 7B gray bars), 50% ethanol for 10 min (Fig 7C black bars), 0.1% sodium hypochlorite (Fig 7C gray bars), and UV radiation in deionized water (Fig 7E) and CDM (Fig 7F). On the other hand, live bacteria were remaining in the suspensions filtered through the 0.45 μm filter in strain Nevada 14 (0.25 CFU), LVS (0.75 CFU), Kato (40.50 CFU), Kf Water (1.25 CFU). Although differing sensitivities were observed among the strains, bacteria were found to survive after detergent treatment (Fig 7D) and UV radiation in undiluted FBS (Fig 7G).

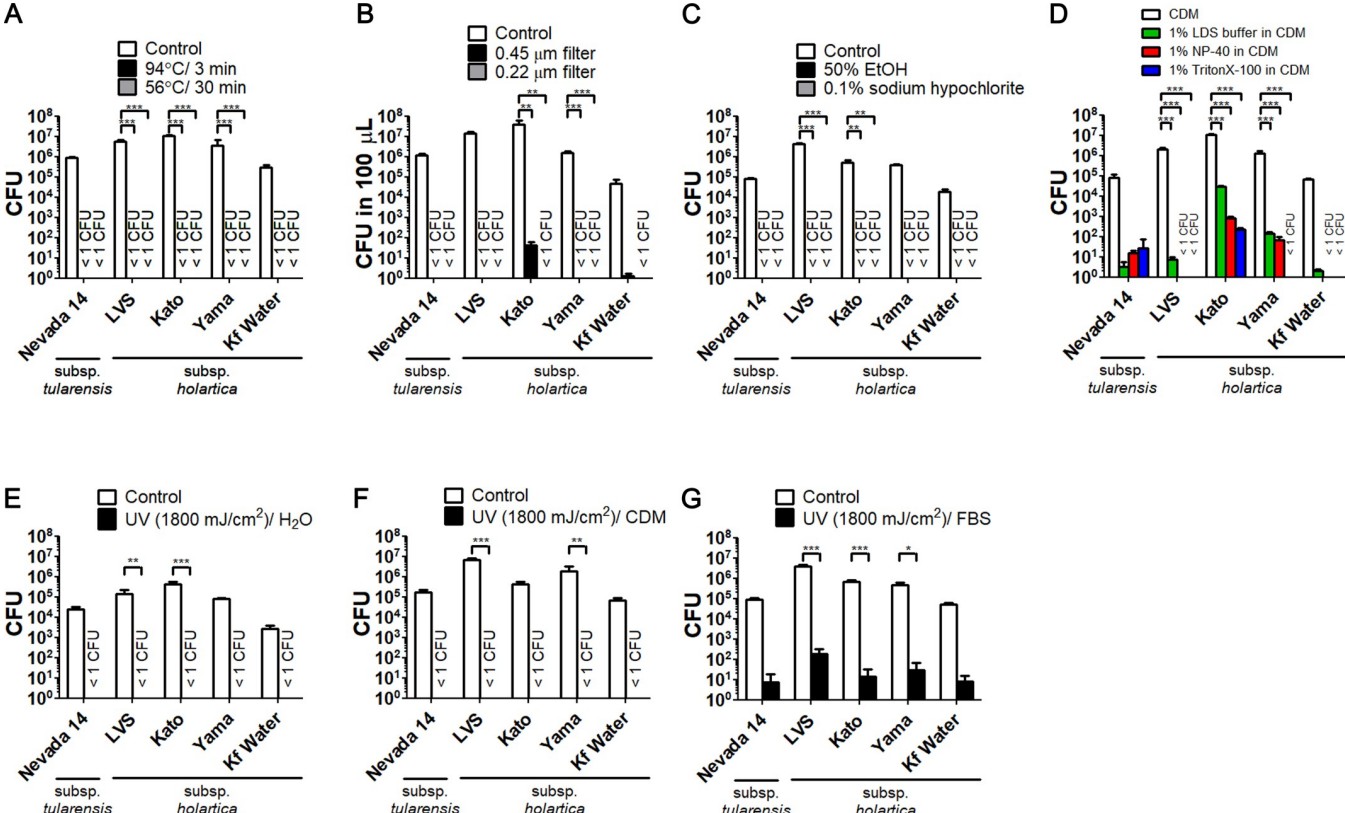

**Fig 7. The validation of effective treatments for the inactivation in five *F. tularensis* strains.** (A) Bacterial suspensions in CDM were heated at 94˚C for 3 min and 56˚C for 30 min and immediately cooled on ice. (B) Bacterial suspensions in 1 mL of CDM were filtered through 0.45 and 0.22 μm pore size membrane filters (Merck Millipore). (C) Bacteria spiked into detergents (1% LDS buffer, 1% NP-40 and 1% TritonX-100) were incubated at 4˚C for 1 day. (E-F) Bacteria suspended in deionized water (E), CDM (F), and undiluted FBS (G) were aliquoted into four 0.2 mL PCR tubes. These samples were simultaneously radiated with UV light. All treated samples were compared with the control samples. All experiments were performed in four replicates. The mean CFU ± SD are shown.

## Discussion

*F. tularensis* is a Gram-negative, non-spore-forming, highly pathogenic and an intracellular bacterium. Samples infected with *F. tularensis* must be inactivated completely in a BSL-3 facility before transportation to a laboratory with a lower BSL for analyses of the bacteria and infected cells. The results of the present study showed that *F. tularensis* SCHU P9 was readily and easily inactivated by heat treatments (94˚C for 3 min and 56˚C for 30 min), filtration using a 0.22 μm filter, and treatments with 70% ethanol, methanol, acetone, 10% neutral buffered formalin and 4% PFA solution. On the other hand, filtration using a 0.45 μm filter and treatments of detergents could significantly decrease the amount of live *F. tularensis* SCHU P9 but were insufficient for the complete removal of bacteria. *F. tularensis* SCHU P9 suspended with undiluted FBS in plastic tubes was resistant to UV radiation but not to deionized water and CDM. The findings of the present study were similar to the common conditions for bacterial inactivation [34]. Here, the actual values of the inactivating condition of *F. tularensis* are provided in detail.

It is known that *F. tularensis* can survive for long periods in soil, fodder, live ticks, animal carcasses, and laboratory culture media [35]. *F. tularensis* is also stable in water and waterborne outbreaks of tularemia caused by *F. tularensis* subsp. *holarctica* has been reported in Bulgaria, Georgia, Germany, Italy, Kosovo, Norway, Russia, Sweden, Turkey, the Czech Republic and the Republic of Georgia [36, 37]. *Francisella noatunensis* NCIMB14265, which was isolated from diseased Atlantic cod held in seawater in Norway [38], was suspended in sterilized seawater microcosms and then culturable bacteria were detected by 40 days at 4˚C [39]. In this study, high concentrations of viable bacteria were detected in CDM for up to 10 weeks, and the viability of *F. tularensis* SCHU P9 suspended in deionized water and undiluted FBS was confirmed at 8 weeks at 4˚C (Fig 1). These data indicate that *F. tularensis* remains viable in CDM and at low temperatures, in agreement with the findings of a previous study [39]. In addition, incubation within 1 h did not affect the viability of *F. tularensis* SCHU P9, as the CFU number did not significantly change from 0 min to 1 h in deionized water, CDM and PBS (Fig 1).

Sera and plasma samples collected from patients and animals are often used for serological diagnosis by detection of specific antibodies with the agglutination test, enzyme-linked immunosorbent assay, indirect immunofluorescence and western blot analysis. To inactivate complement, samples are generally incubated at 56˚C for 30 min. As is widely known, many viruses and non-sporulating bacteria are usually inactivated at 50˚C–60˚C, with the exception of enterococci [40, 41] and thermophilic bacteria. In this study, no bacteria were detected in CDM and undiluted FBS following heat treatment at 56˚C from 10 to 30 min (S1 Fig. and Fig 2B). Moreover, bacterial suspensions in deionized water were quickly inactivated at 56˚C within 1 min. Day *et al.* reported that the time required to reduce the population of *F. tularensis* LVS by 90% (D10-values) in liquid infant formula, apple juice, mango juice, and orange juice was between 8 and 16 s [42]. In this study, the D10-values for *F. tularensis* SCHU P9 in CDM, undiluted FBS, and PBS were 40, 40 and 35 s, respectively. Though the efficiency of the inactivation between our findings and previous data seemed to differ, *Francisella* viability after heat treatment was strongly affected by the solution composition. According to these data, heat treatment at 56˚C for 30 min was sufficient to inactivate *F. tularensis*.

Semipermeable membrane filters with average pore sizes of 0.45 and 0.22 μm are often used to remove bacteria, fungi, cells, aggregated proteins, and debris in liquid samples at microbiological and biomedical laboratories. The separation based on the pore size of the membrane using the filter membrane is a physical removal/separation mechanism and not chemical inactivation. *F. tularensis* with a diameter ranging 0.2–0.7 μm [43] might pass through the

membrane filter having average pore sizes of 0.45 and 0.22 μm and can survive in the filter devices during filtration. In addition, the filter membrane can easily clog if the sample fluid contains abundant aggregated bacteria larger than the pore size of the filter. It is possible that even little pressure can accidentally uncouple the connection between the syringe and the filter. Therefore, the leaked bacteria after filtration and the splashed bacteria during filtration should be paid attention to when filtration methods are applied.

Using 70% ethanol spray is a simple aseptic technique to efficiently inactivate non-sporulating bacteria, but not bacterial spores [44]. However, the most effective concentration of ethanol for the wide spectrum of microbes is 60%–70% [45]. *F. tularensis* SCHU P9 was able to resist deactivation in a solution of 50% ethanol for at least 1 min, although the viability was significantly reduced (Fig 2F). When attempting to remove *F. tularensis* using 70% ethanol solution, the reduction in the ethanol concentration along with the incubation time should be considered.

Depending on the wavelength, UV radiation is classified as UVA (315–400 nm), UVB (280–315 nm) or UVC (200–280 nm). UVC radiation can effectively induce mutations and death of bacterial as well as mammal cells [46]. In particular, UV light at 254 nm, which is near the maximum absorbance in DNA [47], can induce the accumulation of dimers between adjacent thymidine residues in the same DNA strand [33]. Rose *et al*. reported that *F. tularensis* LVS and NY98 suspended in distilled water were inactivated by UVC treatment at 4 mJ/cm$^2$ [48]. We confirmed that *F. tularensis* SCHU P9 spread onto Eugon chocolate agar was completely inactivated by UV radiation at 6.3 mJ/cm$^2$. In addition, bacteria suspended in deionized water and CDM into 1.5 mL tubes and 0.2 mL PCR tube were easily inactivated by UV radiation (Fig 6A). On the other hand, inactivation of bacterial suspensions in undiluted FBS was very difficult with UV radiation (Fig 6B), which suggested that UV radiation failed to inhibit bacteria in the solution containing abundant nutrients, including proteins. In *Bacillus thuringiensis* and *B. anthracis* spores, effective inactivation by UV radiation was also inhibited under more nutritive germination conditions [49]. It is probable that UV radiation is unable to inactivate bacteria because the solution containing abundant nutrients absorbs the UV light into its compounds [50–52].

*F. tularensis* subsp. *tularensis* SCHU P9 was used as a model of the inactivation for *F. tularensis* in this study. The parental strain of *F. tularensis* SCHU P9 is a virulent strain that was initially isolated by Foshay from an ulcer of American patient in 1941 [53, 54]. The Ohara Research Laboratory (Ohara General Hospital, Fukushima, Japan) obtained strain SCHU from the Rocky Mountain Laboratory of the National Institute of Allergy and Infectious Diseases (Hamilton, MT, USA) in 1958 [55]. Afterward, *F. tularensis* SCHU was attenuated by 373 passages in artificial media over 30 years in Japan [55]. Virulent *F. tularensis* SCHU P9 was isolated by *in vivo* passages from attenuated SCHU, as described in a previous report [32]. We believe the methods for inactivation of *F. tularensis* SCHU P9 described in this study would be applicable to other strains of *Francisella* because the viabilities of the treated samples of *F. tularensis* SCHU P9 and the other five strains were similar (Fig 7).

Actual data for the inactivation of *F. tularensis* SCHU P9 are presented in this report. However, it should be noted that the experiments were performed using bacterial suspensions. If the bacteria aggregate in the samples, the conditions required for complete inactivation might differ. Our data are useful for the development of effective inactivation procedures; however, our data cannot be used to confirm the success of other inactivation procedures. We recommend that this study be used as a guide and that each laboratory should validate these procedures in their own laboratory. This is in accordance with current US and Japan law and provides an appropriate perspective of the data reported here. Although there is always a risk of infection of laboratory workers handling *F. tularensis*, the risk can be controlled by complying with the safety protocols.

## Supporting information

**S1 Fig. The viability of F. tularensis SCHU P9 after heated at 56˚C for 10 min.** Bacterial suspensions were prepared with deionized water, CDM, PBS, and undiluted FBS. The samples were heated at 56˚C for 10 min and then immediately cooled on ice. The black and white bars indicate the CFU numbers of the treated and control samples, respectively. Statistical significance was determined by two-way ANOVA with a *post hoc* test (**$p < 0.01$ and ***$p < 0.001$). (TIF)

## Acknowledgments

We are grateful to Dr. Hiromi Fujita for kindly supplying *F. tularensis* subsp. *tularensis* Nevada 14 and subsp. *holarctica* LVS, Kato, Yama, and Kf Water. We also thank Dr. Osamu Fujita for bacterial storage management. We gratefully thank Dr. Tanabayashi and the staff of the Division of Biosafety Control and Research for their excellent technical assistance. The authors would like to thank Enago (www.enago.jp) for the English language review.

## Author Contributions

**Conceptualization:** Mika Azaki, Akihiko Uda, Katsuyoshi Nakazato, Yasuhiro Kawai, Ken Maeda, Shigeru Morikawa.

**Data curation:** Mika Azaki, Akihiko Uda, Akitoyo Hotta, Keita Ishijima, Yudai Kuroda.

**Formal analysis:** Akihiko Uda.

**Funding acquisition:** Akihiko Uda, Shigeru Morikawa.

**Investigation:** Mika Azaki, Akihiko Uda, Akitoyo Hotta, Keita Ishijima, Yudai Kuroda.

**Methodology:** Mika Azaki, Akihiko Uda.

**Project administration:** Akihiko Uda, Shigeru Morikawa.

**Resources:** Akihiko Uda, Akitoyo Hotta.

**Supervision:** Deyu Tian, Katsuyoshi Nakazato, Yasuhiro Kawai, Ken Maeda.

**Validation:** Akihiko Uda.

**Writing – original draft:** Mika Azaki, Akihiko Uda, Deyu Tian, Katsuyoshi Nakazato, Yasuhiro Kawai, Ken Maeda, Shigeru Morikawa.

**Writing – review & editing:** Akihiko Uda, Katsuyoshi Nakazato, Yasuhiro Kawai, Ken Maeda, Shigeru Morikawa.

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
