## [Decision Letter · Decision Letter 0]

28 Sep 2019

PONE-D-19-24860

Effective treatments for the inactivation of Francisella tularensis

PLOS ONE

Dear Dr. Uda,

Thank you for submitting your manuscript to PLOS ONE. After careful consideration, we feel that it has merit but does not fully meet PLOS ONE’s publication criteria as it currently stands. Therefore, we invite you to submit a revised version of the manuscript that addresses the points raised during the review process.

We would appreciate receiving your revised manuscript by Nov 12 2019 11:59PM. To enhance the reproducibility of your results, we recommend that if applicable you deposit your laboratory protocols in protocols.io, where a protocol can be assigned its own identifier (DOI) such that it can be cited independently in the future. For instructions see: http://journals.plos.org/plosone/s/submission-guidelines#loc-laboratory-protocols

We look forward to receiving your revised manuscript.

Kind regards,

Paulo Lee Ho, Ph.D.

Academic Editor

PLOS ONE

Journal Requirements:

Reviewers' comments:

Reviewer's Responses to Questions

**Comments to the Author**

1. Is the manuscript technically sound, and do the data support the conclusions?

Reviewer #1: Yes

Reviewer #2: Yes

2. Has the statistical analysis been performed appropriately and rigorously? 

Reviewer #1: Yes

Reviewer #2: Yes

3. Have the authors made all data underlying the findings in their manuscript fully available?

Reviewer #1: Yes

Reviewer #2: Yes

4. Is the manuscript presented in an intelligible fashion and written in standard English?

Reviewer #1: Yes

Reviewer #2: Yes

5. Review Comments to the Author

Reviewer #1: The manuscript entitled “effective treatments for the inactivation of Francisella tularensis” by Azaki et al. is extremely straight forward, easy to follow, and for the most part well written. I would require several edits and clarifications prior to publication:

1. Consider changing title to “Effective methods for the inactivation of Francisella tularensis”

2. Abstract line 34. Change to read “…ultraviolet radiation compared to suspensions…” and change throughout manuscript, irradiation is the act of exposing to radiation, whereas ultraviolet light is a form of radiation.

3. Abstract line 37. What is meant by “conventional conditions required”? This sentence is confusing to the reviewer because most of the processes described in the manuscript would be considered quite conventional.

4. Page 3 line 44. The statement regarding LAI in the United States prior to 1978 is misleading. This is true particularly in context of the first sentence of the introduction. While I do not disagree that human error is the primary reason for LAI, the data provided prior to 1978, largely did not include engineering controls such as biosafety cabinets and respiratory protection. Please consider revising this first paragraph for clarity.

5. Page 3 line 50. Change to read “…have not been clearly identified…”

6. Page 4. General comment. To make your work more applicable to international standards, I would also briefly discuss the BMBL –Biosafety in Microbiological and Biomedical Laboratories Guide. This document is used to govern most or all US labs. Additionally, the goal of this paper is to demonstrate the validation of inactivation of Tier one select agents, this is now required by US law. This law is briefly described and referenced in Chua et al. https://wwwnc.cdc.gov/eid/article/25/5/18-0928_article. This paper should also be cited in the current manuscript as it discusses the use of formaldehyde for inactivating F. tularensis.

7. Page 5 line 104. Change to read “Therefore, the present study aimed to confirm….”

8. Page 12 and applicable to throughout the manuscript, consider making a statement in the method section (if it is there and I missed it, I apologize) that all FBS was used at 10% final concentration (if that is the case).

9. Page 13. Line 242, change “resistance of sterilization” to “effectiveness of filter sterilization”

10. Page 10 line 248 and Page 20 Line 368. While I do not disagree that 70% ethanol is effective, I do disagree that 70% ethanol is “the most effective”.

11. Page 16 line 306. Change “had failed” to “was not achieved”

12. Page 17 line 322. Change to read “…bacteria were found…” as bacteria is plural.

13. Page 18 line 338. This first sentence warrants a reference being cited, perhaps a review article.

14. Page 20. Starting on line 375. The authors should postulate why the addition of the FBS hindered inactivation but CDM did not. A publication (Omotade et al. J Applied Microbio https://doi.org/10.1111/jam.12644) could be cited and discussed because although this work uses B. anthracis spores it does discuss UV interference for bacterial inactivation, even in the presence of minimal compounds.

15. Page 20 and 21. Lines 308-399. I believe these details are important but the strain origins and characteristics would be better presented in a table format.

16. Page 21. Line 406. I would recommend the authors make a statement indicating that this work should be used as a guide and each laboratory should validate these procedures in their own laboratory. This would be in accordance with current US Law but also would provide appropriate perspective of the data reported here.

17. Page 27 line 579. Change “stood” to “maintained”

18. Figures and throughout. The term “untreatment” is not commonly used, I would change to “control” or “no treatment”

Reviewer #2: Review of PONE-D-19-24860

Effective Treatments for the Inactivation of Francisella tularensis

This manuscript is a well-written summary of experiments designed to determine the inactivation efficacy of numerous approaches for Francisella tularensis. The data are useful to laboratorians working with this organism and will improve biosafety procedures within BSL-3 laboratories.

Specific Comments:

Line 34 – missing word in sentence. “…highly resistant to ultraviolet irradiation than suspensions…” Suggest changing to “…highly resistant to ultraviolet irradiation, more so than suspensions…”

Line 41-42 – reference for this statement (human error is leading cause) is needed

Line 68 – no change needed, but the BMBL is an additional and excellent reference to take note of

https://www.aaalac.org/accreditation/RefResources/BMBL.pdf

Line 94-103 – Calfee and Wendling (Science of the Total Environment, 2013) is another reference for previously studied inactivation methods for Ft. This study may be worth including as a reference for further background on non-suspension-type inactivation studies.

https://www.ncbi.nlm.nih.gov/pubmed/23208274

Line 110 – “CDM” first use, please define

Line 124 – “FBS” first use, please define

Line 142 – please add/indicate the filter media type (PES, MCE, etc.) and the supplier part number

Line 152-153 – Statement is confusing, check for accuracy. “After measuring the CFU in the supernatant and pellets, the samples were serially diluted and cultured…” were viable cells (CFU determination) determined for pellets and supers separately? Then combined and determined? Please add clarification.

Line 177 – Please replace the ‘ in 1/4’ with actual units. Please use metric units when possible.

Line 196 – please indicate filter type and part#

Line 248 – suggest deletion of first sentence in this paragraph. “most effective agent” is subjective. Effectiveness depends on many factors (surface type contaminated, biological agent, temperature, humidity, organic load, etc.), therefore one ‘agent’ cannot simply be labeled the ‘most effective’ for all situations. It may be the most ‘common’ disinfectant in laboratories, but one could argue that iodine or sodium hypochlorite are superior disinfectants.

Line 301- First sentence of this paragraph is written as though the mechanism of inactivation was determined in this study (thymine dimer generation), however, this is not the case. Suggest deleting this first sentence, or rewording to indicate that UV light is reported to inactivate bacterial cells via thymine dimer formation, and provide reference.

Line 330 – Here and throughout, filters are described no differently as chemical and physical inactivation methods. However, filtration is a physical removal/separation mechanism, and likely inactivates very few organisms. Suggest adding some text to clarify this difference here, and potentially other areas in the manuscript where filtration is mentioned.

Line 333 – mention of “complete inactivation”, see comment above about filtration and inactivation versus separation

Line 368 – see previous comment about ethanol as the ‘most effective’ technique to inactivate non-spore-forming bacteria

Line 403 – Suggest revision of sentence suggesting “our data should be useful to confirm whether samples are inactivated”. I suggest that your data is useful to develop effective inactivation procedures, however you data cannot be used to confirm others’ inactivation procedures success.

6. PLOS authors have the option to publish the peer review history of their article (what does this mean?). If published, this will include your full peer review and any attached files.

Reviewer #1: No

Reviewer #2: No

---

## [Author Response · Author response to Decision Letter 0]

24 Oct 2019

Dear Prof. Paulo Lee Ho:

Thank you for your response dated September 28, 2019 to our manuscript (PONE-D-19-24860; Effective methods for the inactivation of Francisella tularensis). We have revised our manuscript according to the reviewers’ comments. We hope that the revised manuscript will be considered for publication in PLOS ONE. 

Thank you for your kind consideration.

Our responses to the reviewers’ comments are as follows:

Journal Requirements:

1. When submitting your revision, we need you to address these additional requirements. Please ensure that your manuscript meets PLOS ONE's style requirements, including those for file naming. The PLOS ONE style templates can be found at

We have corrected the manuscript and figure file names to meet PLOS ONE’s style requirements.

We have shown all data by adding Fig. 5 and S1 Figure in the revised manuscript according to the reviewer’s suggestion.

Reviewer #1: 

1. Consider changing title to “Effective methods for the inactivation of Francisella tularensis”

As per your suggestion, we have revised the title.

2. Abstract line 34. Change to read “…ultraviolet radiation compared to suspensions…” and change throughout manuscript, irradiation is the act of exposing to radiation, whereas ultraviolet light is a form of radiation.

According to your comment, we have replaced “irradiation” with “radiation” in the revised manuscript.

3. Abstract line 37. What is meant by “conventional conditions required”? This sentence is confusing to the reviewer because most of the processes described in the manuscript would be considered quite conventional.

To avoid confusion, we have deleted the sentence regarding “conventional conditions required” and have added the following sentence to the revised manuscript (page 2 lines 53–55).

The data presented in this study could be useful for the establishment of guidelines and standard operating procedures (SOP) to inactivate the contaminated samples in not only F. tularensis but also other bacteria.

4. Page 3 line 44. The statement regarding LAI in the United States prior to 1978 is misleading. This is true particularly in context of the first sentence of the introduction. While I do not disagree that human error is the primary reason for LAI, the data provided prior to 1978, largely did not include engineering controls such as biosafety cabinets and respiratory protection. Please consider revising this first paragraph for clarity.

As suggested by the reviewer, we have modified the first sentence of the Introduction section (page 3 lines 59–61) as follows:

Laboratory-acquired infections (LAIs) are caused by accidental exposure to infectious aerosols and contact with mucous membranes, even though LAIs have been decreased due to personal protective measures and biosafety training [1, 2].

5. Page 3 line 50. Change to read “…have not been clearly identified…”

We have corrected the sentence according to the reviewer’s comment in the revised manuscript (page 3 line 69).

6. Page 4. General comment. To make your work more applicable to international standards, I would also briefly discuss the BMBL –Biosafety in Microbiological and Biomedical Laboratories Guide. This document is used to govern most or all US labs. Additionally, the goal of this paper is to demonstrate the validation of inactivation of Tier one select agents, this is now required by US law. This law is briefly described and referenced in Chua et al. https://wwwnc.cdc.gov/eid/article/25/5/18-0928_article. This paper should also be cited in the current manuscript as it discusses the use of formaldehyde for inactivating F. tularensis.

We have added the relevant sentences regarding the Biosafety in Microbiological and Biomedical Laboratories (BMBL) Guide (page 4 line 90) and reference #26 (page 5 line 119–120) in the revised manuscript.

7. Page 5 line 104. Change to read “Therefore, the present study aimed to confirm….”

We have corrected the sentence according to the reviewer’s comment (page 6 line 130).

8. Page 12 and applicable to throughout the manuscript, consider making a statement in the method section (if it is there and I missed it, I apologize) that all FBS was used at 10% final concentration (if that is the case).

We have used “undiluted FBS” in this study. I have replaced “FBS” with “undiluted FBS” in order to avoid confusion of the final concentration used.

9. Page 13. Line 242, change “resistance of sterilization” to “effectiveness of filter sterilization”

We have corrected the sentence according to the reviewer’s comment (page 14 line 288).

10. Page 10 line 248 and Page 20 Line 368. While I do not disagree that 70% ethanol is effective, I do disagree that 70% ethanol is “the most effective”.

As per the reviewer’s suggestion, we have revised in the statement as follows (page 14 line 295 to 296):

Using a solution of 70% ethanol is a simple aseptic technique to inactivate pathogens.

11. Page 16 line 306. Change “had failed” to “was not achieved”

We have corrected the sentence according to the reviewer’s comment (page 17 line 354).

12. Page 17 line 322. Change to read “…bacteria were found…” as bacteria is plural.

We have corrected the sentence according to the reviewer’s comment (page 18 line 371).

13. Page 18 line 338. This first sentence warrants a reference being cited, perhaps a review article.

As per the reviewer’s comment, we have cited the relevant reference in the revised manuscript (page 19 line 388).

14. Page 20. Starting on line 375. The authors should postulate why the addition of the FBS hindered inactivation but CDM did not. A publication (Omotade et al. J Applied Microbio https://doi.org/10.1111/jam.12644) could be cited and discussed because although this work uses B. anthracis spores it does discuss UV interference for bacterial inactivation, even in the presence of minimal compounds.

The following sentences have been added (page 22 line 448–452).

In Bacillus thuringiensis and B. anthracis spores, effective inactivation by UV radiation was also inhibited under more nutritive germination conditions [47]. It is probable that UV radiation is unable to inactivate bacteria because the solution containing abundant nutrients absorbs the UV light into its compounds [48-50].

15. Page 20 and 21. Lines 308-399. I believe these details are important but the strain origins and characteristics would be better presented in a table format.

The strains used in this study have been summarized in Table 1 (page 7 line 145).

16. Page 21. Line 406. I would recommend the authors make a statement indicating that this work should be used as a guide and each laboratory should validate these procedures in their own laboratory. This would be in accordance with current US Law but also would provide appropriate perspective of the data reported here.

The following sentences have been added in the revised manuscript (page 23 lines 470–472).

We recommend that this study be used as a guide and that each laboratory should validate these procedures in their own laboratory. This is in accordance with current US and Japan law and provides an appropriate perspective of the data reported here.

17. Page 27 line 579. Change “stood” to “maintained”

As per the reviewer’ comment, I corrected this in the revised manuscript (page 31 line 686).

18. Figures and throughout. The term “untreatment” is not commonly used, I would change to “control” or “no treatment”

According to the reviewer’s comment, “untreatment” has been replaced with “control” in the revised manuscript.

Reviewer #2:

1. Line 34 – missing word in sentence. “…highly resistant to ultraviolet irradiation than suspensions…” Suggest changing to “…highly resistant to ultraviolet irradiation, more so than suspensions…”

Please refer to response 2 provided to Reviewer 1.

2. Line 41-42 – reference for this statement (human error is leading cause) is needed.

Please refer to reply 4 provided to Reviewer 1.

3. Line 68 – no change needed, but the BMBL is an additional and excellent reference to take note of https://www.aaalac.org/accreditation/RefResources/BMBL.pdf

Please refer to response 6 provided to Reviewer 1.

4. Line 94-103 – Calfee and Wendling (Science of the Total Environment, 2013) is another reference for previously studied inactivation methods for Ft. This study may be worth including as a reference for further background on non-suspension-type inactivation studies. https://www.ncbi.nlm.nih.gov/pubmed/23208274

We have cited the specified reference in the Introduction section as reference #30 in the revised manuscript (page 6 line 125).

5. Line 110 – “CDM” first use, please define

We have defined CDM at its specified instance in the revised manuscript (page 7 line 136).

6. Line 124 – “FBS” first use, please define

We have defined FBS at the specified instance in the revised manuscript (page 7 line 149).

7. Line 142 – please add/indicate the filter media type (PES, MCE, etc.) and the supplier part number

We have added the filter media type and supplier part number in the revised manuscript (page 8 lines 167–169).

8. Line 152-153 – Statement is confusing, check for accuracy. “After measuring the CFU in the supernatant and pellets, the samples were serially diluted and cultured…” were viable cells (CFU determination) determined for pellets and supers separately? Then combined and determined? Please add clarification.

We have deleted this incorrect sentence in the revised manuscript.

9. Line 177 – Please replace the ‘ in 1/4’ with actual units. Please use metric units when possible.

We have replaced “in ¼” with “in 6.35 mm” in the revised manuscript (page 10 line 202).

10. Line 196 – please indicate filter type and part#

We have added the filter media type and supplier part number in the revised manuscript (page 11 lines 239–241).

11. Line 248 – suggest deletion of first sentence in this paragraph. “most effective agent” is subjective. Effectiveness depends on many factors (surface type contaminated, biological agent, temperature, humidity, organic load, etc.), therefore one ‘agent’ cannot simply be labeled the ‘most effective’ for all situations. It may be the most ‘common’ disinfectant in laboratories, but one could argue that iodine or sodium hypochlorite are superior disinfectants.

Please refer to response #10 provided to Reviewer 1.

12. Line 301- First sentence of this paragraph is written as though the mechanism of inactivation was determined in this study (thymine dimer generation), however, this is not the case. Suggest deleting this first sentence, or rewording to indicate that UV light is reported to inactivate bacterial cells via thymine dimer formation, and provide reference.

According to the reviewer’s comment, we have corrected the sentence and have added a corresponding reference in the revised manuscript (page 17 line 350).

13. Line 330 – Here and throughout, filters are described no differently as chemical and physical inactivation methods. However, filtration is a physical removal/separation mechanism, and likely inactivates very few organisms. Suggest adding some text to clarify this difference here, and potentially other areas in the manuscript where filtration is mentioned.

Based on the reviewer’s comment, we have discussed that filtration was a physical removal/separation mechanism and most likely inactivates very few organisms in the revised manuscript as follows (page 21 lines 417–428): 

Semipermeable membrane filters with average pore sizes of 0.45 and 0.22 μm are often used to remove bacteria, fungi, cells, aggregated proteins, and debris in liquid samples at microbiological and biomedical laboratories. The separation based on the pore size of the membrane using the filter membrane is a physical removal/separation mechanism and not chemical inactivation. F. tularensis with a diameter ranging 0.2–0.7 μm [42] might pass through the membrane filter having average pore sizes of 0.45 and 0.22 μm and can survive in the filter devices during filtration. In addition, the filter membrane can easily clog if the sample fluid contains abundant aggregated bacteria larger than the pore size of the filter. It is possible that even little pressure can accidentally uncouple the connection between the syringe and the filter. Therefore, the leaked bacteria after filtration and the splashed bacteria during filtration should be paid attention to when filtration methods are applied.

14. Line 333 – mention of “complete inactivation”, see comment above about filtration and inactivation versus separation

According to the reviewer’s comment, we have replaced “complete inactivation” with “the complete removal of bacteria” in the revised manuscript (page 19 line 382).

15. Line 368 – see previous comment about ethanol as the ‘most effective’ technique to inactivate non-spore-forming bacteria

According to the reviewer’s comment, we have corrected this sentence and added a corresponding reference in the revised manuscript (page 21 lines 429–430) as follows:

Using 70% ethanol spray is a simple aseptic technique to efficiently inactivate non-sporulating bacteria but not bacterial spores (32).

16. Line 403 – Suggest revision of sentence suggesting “our data should be useful to confirm whether samples are inactivated”. I suggest that your data is useful to develop effective inactivation procedures, however you data cannot be used to confirm others’ inactivation procedures success.

According to the reviewer’s comment, we have revised the sentence and have added a corresponding reference in the revised manuscript (page 23 lines 468–470) as follows:

Our data are useful for the development of effective inactivation procedures; however, our data cannot be used to confirm the success of other inactivation procedures.

---

## [Decision Letter · Decision Letter 1]

31 Oct 2019

Effective methods for the inactivation of Francisella tularensis

PONE-D-19-24860R1

Dear Dr. Uda,

We are pleased to inform you that your manuscript has been judged scientifically suitable for publication and will be formally accepted for publication once it complies with all outstanding technical requirements.

With kind regards,

Paulo Lee Ho, Ph.D.

Academic Editor

PLOS ONE

Additional Editor Comments (optional):

Reviewers' comments:

Reviewer's Responses to Questions

**Comments to the Author**

1. If the authors have adequately addressed your comments raised in a previous round of review and you feel that this manuscript is now acceptable for publication, you may indicate that here to bypass the “Comments to the Author” section, enter your conflict of interest statement in the “Confidential to Editor” section, and submit your "Accept" recommendation.

Reviewer #1: All comments have been addressed

Reviewer #2: All comments have been addressed

2. Is the manuscript technically sound, and do the data support the conclusions?

Reviewer #1: Yes

Reviewer #2: Yes

3. Has the statistical analysis been performed appropriately and rigorously? 

Reviewer #1: Yes

Reviewer #2: Yes

4. Have the authors made all data underlying the findings in their manuscript fully available?

Reviewer #1: Yes

Reviewer #2: Yes

5. Is the manuscript presented in an intelligible fashion and written in standard English?

Reviewer #1: Yes

Reviewer #2: Yes

6. Review Comments to the Author

Reviewer #1: much improved manuscript and thanks to the authors for addressing my comments from the original review****

Reviewer #2: (No Response)

7. PLOS authors have the option to publish the peer review history of their article (what does this mean?). If published, this will include your full peer review and any attached files.

Reviewer #1: No

Reviewer #2: No

---

## [Editor Report · Acceptance letter]

6 Nov 2019

PONE-D-19-24860R1 

Effective methods for the inactivation of Francisella tularensis 

Dear Dr. Uda:

I am pleased to inform you that your manuscript has been deemed suitable for publication in PLOS ONE. Congratulations! Your manuscript is now with our production department. 

With kind regards,

on behalf of

Dr. Paulo Lee Ho 

Academic Editor

PLOS ONE